# Zika virus-like particle vaccine protects AG129 mice and rhesus macaques against Zika virus

**Lo Vang**[1]*, **Christopher S. Morello**[1], **Jason Mendy**[1], **Danielle Thompson**[1], **Darly Manayani**[2¤a], **Ben Guenther**[1], **Justin Julander**[3], **Daniel Sanford**[4], **Amit Jain**[1], **Amish Patel**[1], **Paul Shabram**[1], **Jonathan Smith**[2¤b], **Jeff Alexander**[1,2]

**1** Emergent BioSolutions Inc., Gaithersburg, Maryland, United States of America, **2** PaxVax Inc., San Diego, California, United States of America (PaxVax was acquired by Emergent BioSolutions Inc. Oct 2018), **3** Institute for Antiviral Research, Department of Animal, Dairy, and Veterinary Sciences, Utah State University, Logan, Utah, United States of America, **4** Battelle Biomedical Research Center, West Jefferson, Ohio, United States of America

¤a Current address: Novartis Gene Therapies, San Diego, California, United States of America
¤b Current address: VLP Therapeutics, LLC, Gaithersburg, Maryland, United States of America
* VangL@ebsi.com

**Data Availability Statement:** All relevant data are within the manuscript and its Supporting Information files.

## Abstract

### Background

Zika virus (ZIKV), a mosquito-borne flavivirus, is a re-emerging virus that constitutes a public health threat due to its recent global spread, recurrent outbreaks, and infections that are associated with neurological abnormalities in developing fetuses and Guillain-Barré syndrome in adults. To date, there are no approved vaccines against ZIKV infection. Various preclinical and clinical development programs are currently ongoing in an effort to bring forward a vaccine for ZIKV.

### Methodology/Principle findings

We have developed a ZIKV vaccine candidate based on Virus-Like-Particles (VLPs) produced in HEK293 mammalian cells using the prM (a precursor to M protein) and envelope (E) structural protein genes from ZIKV. Transient transfection of cells via plasmid and electroporation produced VLPs which were subsequently purified by column chromatography yielding approximately 2mg/L. Initially, immunogenicity and efficacy were evaluated in AG129 mice using a dose titration of VLP with and without Alhydrogel 2% (alum) adjuvant. We found that VLP with and without alum elicited ZIKV-specific serum neutralizing antibodies (nAbs) and that titers correlated with protection. A follow-up immunogenicity and efficacy study in rhesus macaques was performed using VLP formulated with alum. Multiple neutralization assay methods were performed on immune sera including a plaque reduction neutralization test, a microneutralization assay, and a Zika virus *Renilla* luciferase neutralization assay. All of these assays indicate that following immunization, VLP induces high titer nAbs which correlate with protection against ZIKV challenge.

### Conclusions/Significance

These studies confirm that ZIKV VLPs could be efficiently generated and purified. Upon VLP immunization, in both mice and NHPs, nAb was induced that correlate with protection

**Funding:** This study was funded by NIH under contract # HHSN272201700041I/HHSN27200004 to JJ; and by NIH under contract #s HHSN272201200003I/HHSN27200026 and HHSN272201200003I/ HHSN27200019 to DS. The funders were involved in study design but had no role in data collection and analysis, decision to publish, or preparation of the manuscript.

**Competing interests:** I have read the journal's policy and the authors of this manuscript have the following competing interests: Jonathan Smith and Darly Manayani have no competing interests. Jeff Alexander is a paid consultant of Emergent. Justin Julander and Daniel Sanford are paid employees of Utah State University and Battelle Biomedical Research Center, respectively. Lo Vang, Chris Morello, Jason Mendy, Danielle Thompson, Ben Guenther, Amit Jain, Amish Patel, and Paul Shabram are paid employees of Emergent BioSolutions Inc.

against ZIKV challenge. These studies support translational efforts in developing a ZIKV VLP vaccine for evaluation in human clinical trials.

## Author summary

Zika virus (ZIKV) is a significant global health threat particularly due to the speed in which epidemics can occur. The resulting infections have been demonstrated to harm a developing fetus and, in some adults, be a co-factor for the development of Guillain-Barré syndrome. ZIKV is typically spread by the *Aedes* mosquito, but sexual transmission is also possible. We sought to develop a ZIKV prophylactic vaccine based on surface glycoproteins of the virus that would be devoid of any viral genetic material. This Virus-Like-Particle (VLP) was generated *in vitro* following introduction of plasmid DNA encoding Zika structural protein (prM-E) genes into mammalian cells. The aluminum-adjuvanted VLP induced nAbs in mice and nonhuman primates and protected against ZIKV challenge *in vivo*. These studies support the evaluation of this VLP candidate vaccine in human clinical trials.

## Introduction

Zika virus (ZIKV), a positive-sense RNA virus, belongs to the genus *Flavivirus*, family *Flaviviridae* which includes other disease pathogens of interest such as West Nile, dengue, Tick-borne encephalitis, yellow fever, and Japanese encephalitis viruses [1]. ZIKV was first isolated from a sentinel rhesus macaque in the Zika forest, Uganda in 1947 and was presumed to be limited to the African and Asian tropical zones for several decades [2]. The flaviviruses are transmitted by mosquito or tick vectors with ZIKV being unusual in its capacity to also be spread by sexual contact [3–7]. ZIKV infection was thought to be typically asymptomatic with a minority of subjects presenting mild symptoms including rash, fever, and headache [8].

ZIKV emerged in several major outbreaks of infection first recorded in Yap Island, Micronesia (2007; approximately 73% of 6,892 residents infected with almost 900 symptomatic cases) and then in French Polynesia (2013; approximately 11% of 270,000 infected with approximately 30,000 symptomatic cases) [8–10]. The virus subsequently reached Brazil in 2015 and was further spread to Central and North America in 2016. It was during the 2013 outbreak that Guillain-Barré syndrome was first described in association with ZIKV infection [11,12]. Additional pathologies were observed during ZIKV outbreak in Northeastern Brazil where evidence accumulated supporting a causal relationship between ZIKV infection and fetal injury, including; preterm birth, fetal death and stillbirth, and congenital malformations, including microcephaly, abnormal brain development, limb contractures, eye abnormalities, brain calcifications, and other neurologic abnormalities [13–21]. Incidence of ZIKV in the Americas peaked in 2016 and has since dramatically declined in subsequent years [22].

Licensed flavivirus vaccines are available for yellow fever, Japanese encephalitis, dengue, and tick-borne encephalitis viruses based on inactivated or live-attenuated viruses [23–26]. There is not an approved vaccine for West Nile virus although human vaccines are in development and a vaccine is available for horses [27]. Currently, there are no approved ZIKV vaccines that could prevent viral infection of the mother and transmission to the developing fetus, although several vaccine candidates are in preclinical and clinical development [28–35]. Various traditional vaccine platforms used in ZIKA vaccine development include DNA, viral-

vectored (Measles/Adenovirus/VSV/DENV/MVA) and inactivated virions. The mRNA vaccine approach although 'not traditional' has recently been developed to address COVID-19 disease and has been granted Emergency Use Authorization (Pfizer/BioNTech and Moderna, December 2020). The delivery of VLPs, based on the use of structural proteins as a vaccine, can be achieved by different approaches including DNA (e.g., NIAID/VRC), mRNA (e.g., Moderna), and viral vectors (e.g., Themis Bioscience). We choose to generate the VLPs for use as a vaccine by *in vitro* introduction of plasmid DNA encoding Zika structural protein (prM-E) genes into mammalian cells. We used a chimeric vaccine design whereby the E ectodomain was derived from a recent circulating Brazilian strain SPH2015 and the prM/E stem-anchor sequences were derived from the African MR766 strain. The strategy for the chimeric design was two-fold; promote generation of an immune response to a recently circulating strain and selection of the optimal chimeric design to achieve highest VLP yield *in vitro*. As described in our previous publication [36], we demonstrated that mice immunized with ZIKV VLPs induced immune sera that upon transfer to AG129 mice were protected against ZIKV challenge. Studies described herein continue our VLP vaccine development approach in AG129 mice and rhesus macaque animal models.

During flavivirus replication, the viral genome encodes a polyprotein precursor that is processed into 3 structural proteins [capsid, precursor-to-membrane (prM-approximately 168 amino acids (aa)), envelope (E)] and 7 nonstructural proteins [37]. The immature virion with trimeric spikes of prM-E dimers is cleaved by a furin-like serine protease within the trans-Golgi to produce a mature ZIKV particle with a smooth outer shell made of 90 heterodimers of E (approximately 504 aa) and M (approximately 75 aa) proteins [38–40]. The prM cleavage maybe incomplete, and thus, a virion with both mature and immature regions are released from infected cells [1,39,41]. Antibodies (Abs) specific for prM have been identified following infection [42,43]. Isolation of monoclonal antibodies (mAbs) from ZIKV-infected donors has revealed that most of the humoral immune responses were directed to the E protein [44–48]. The subunits of E protein consist of 3 domains (DI, DII, DIII) where DIII contributes to viral attachment and DII promotes fusion between viral and target cell membrane [39,49,50]. In general, DIII-specific Abs are type specific and neutralizing whereas Abs against DI/II are cross-reactive and poorly neutralizing [47,51]. In addition, tertiary/quaternary epitopes involving either 2 subdomains within the same protomer or 2 identical subdomains from different protomers have been identified as targets of potent nAbs [48,52,53].

We sought to develop a ZIKV vaccine based on VLPs produced in mammalian cells to elicit protective nAb responses, largely directed against the ZIKV E protein. A plasmid encoding the prM-E structural proteins was transiently transfected into HEK293 cells to produce the VLPs. During process development, methods were identified for upstream VLP production and downstream purification that are compatible with large-scale production. Purified VLPs were used to immunize AG129 mice and rhesus macaques wherein immune responses were induced which protected animals against ZIKV challenge. Protection was observed to be correlated with virus-specific nAbs. These data support further ZIKV VLP vaccine development including evaluation in human clinical trials for safety, immunogenicity, and efficacy.

## Materials and methods

### Ethical statements

**Ethics statement for AG129 mice Utah State University.** All AG129 mouse animal procedures were performed at Utah State University, Logan UT. AG129 mice were produced from in-house colonies. Veterinary care and experimental procedures were performed in the AAALAC-accredited Laboratory, Animal Research Center at Utah State University with the

approval of the Institutional Animal Care and Use Committee of Utah State University. Study approval and animal care was conducted in accordance with *The Guide for Care and Use of Laboratory Animals* and U.S. Government Principles for the Utilization and Care of Vertebrate Animals Used in Testing, Research and Training.

**Ethics statement Battelle for rhesus macaques.** Rhesus studies were carried out at Battelle Biomedical Research Center (Battelle), West Jefferson, OH. All animals were cared for in compliance with the *Guide for Care and Use of Laboratory Animals* and procedures were approved by Battelle's IACUC. Rhesus macaques were housed in an AAALAC-accredited facility.

## ZIKV VLP DNA design, production, purification, and characterization of VLPs

The ZIKV cassette plasmid construct used to generate VLP was previously described [36]. The ZIKV cassette used to generate the VLP consists of: 1) the human IL2 signal sequence (MYRMQLLSCIALSLALVTNS) for prM; 2) the prM sequence contains 168 aa, starting with AEI and ending with AYS from the African MR766 strain; 3) the E ectodomain from the Brazilian SPH2015 strain contains 405 aa, starting with IRC and ending with SGS; and 4) the stem-anchor from the African MR766 strain contains 99 aa starting with TIG and ending with VSA. The cassette containing the chimera ZIKV structural genes, prM-E, was inserted downstream from a human CMV IE enhancer/promoter in the CMV/R plasmid described by Barouch *et al.* [54]. The plasmid containing the structural genes was used to electroporate HEK293 cells to produce supernatant containing ZIKV VLPs.

For VLP purification, the clarified harvest was concentrated and diafiltered into load buffer and purified using a two-column chromatography process as described previously [36]. The purified VLPs were buffer exchanged into formulation buffer (25mM TRIS hydrochloride and 25mM sodium citrate dihydrate) and sterile filtered for animal studies. Harvest supernatant containing VLPs, or column purified VLPs, were resolved by NuPAGE 4–12% Bis-Tris precast protein gels (Invitrogen, CA) and stained with InstantBlue Coomassie stain reagent (Expedeon) or transferred onto a nitrocellulose membrane using iBlot dry blotting system (Invitrogen, CA). The nitrocellulose membrane with transferred protein was blocked with 5% milk (Labscientific, Inc., NJ) in phosphate buffered saline containing 0.05% Tween 20 (PBS-T) and incubated with a mouse monoclonal antibody (mAb) (MyBioSource, CA) to recombinant (generated in insect cells) E protein (MyBiosource) at 1:1000 dilution. The membrane was washed 3 times with PBS-T and probed with goat anti-mouse HRP-conjugated antibody (Invitrogen) at 1:10,000 dilution. The western image was developed by using an enhanced chemiluminescent (ECL) substrate kit SuperSignal West Femto (ThermoFisher Scientific). Each batch of VLP was quality tested to confirm identity and purity by performing SDS page, Western Blot, and dynamic light scattering (DLS) analyses.

## ZIKV challenge strain

The ZIKV strain used for challenge studies in AG129 mice was Puerto Rican strain PRVABC59 (BEI Resources, Manassas, VA, USA). It was passaged 2 times in Vero 76 cells. Infected cells and supernatant were frozen once, thawed, centrifuged to remove cell debris, aliquoted and stored at -80˚C for subsequent use in mice.

Virus for NHP challenge utilized a fully characterized working bank of ZIKV strain PRVABC59 (Lot # 4350-WB) at a titer of $1.08 \times 10^6$ PFU/mL. The strain source was ATCC/ BEI Resources [Lot# 64112564 (NR-50240), Manassas, Virginia, Gen Bank: KX087101], where the virus was passaged 3 times (Centers for Disease Control) followed by 2 additional passages (BEI Resources) all in Vero E6 cell lines. One final passage at Battelle was performed at a MOI

of 0.0024, in Vero E6 cell line (Lot 3161-WCB-1 at passage 42), harvested 6 days post-infection with approximately 70% cytopathic effect.

## Immunogenicity and efficacy studies

AG129 mice lack the receptor for types I and II IFN (IFN α/β and ϒ). The mice display age-dependent morbidity and mortality, providing a platform for testing the efficacy of antivirals and vaccines. Six- to 8-week-old AG129 mice (produced from in-house colonies at Utah State University) were immunized with ZIKV VLP (50μL each hind limb, 100μL total) by the intramuscular (IM) route on Day 0 followed by a booster immunization given on Day 28 with doses of 10, 1, or 0.1μg with or without Alhydrogel 2% (alum) adjuvant (Brenntag Biosector A/S, Frederikssund, Denmark) (100μg per dose), constituting Groups 1, 2, 3, 4, 5, and 6. Group 7 were control mice immunized with alum only. Mice were monitored twice daily for the duration of the study and bled on Days 14, and 42 to evaluate induction of ZIKV-specific nAbs. Mice were challenged on Day 44 with ZIKV strain (PRVABC59) with ~100 $CCID_{50}$ delivered by subcutaneous (SC) injection into the inguinal space between the back-right leg and the body in 0.1 mL volume. Individual weights were taken every other day after virus challenge through 28 days post-virus challenge. Mice were bled on day 5 post-challenge (Day 49) to quantify ZIKV relative genomic RNA copies by RT-qPCR. Mice were monitored twice daily for 28 days post-challenge and mice that were moribund or lost greater than 20% of their starting weight were humanely euthanized. Average survival time, percent mortality, and weight changes were evaluated.

All rhesus macaques were procured from Covance Research Products (Alice, TX) and studies carried out at Battelle Biomedical Research Center, West Jefferson, OH. A total of 22 (11 male and 11 female) Chinese-origin rhesus macaques (*Macaca mulatta*), weighing 3.8–6.3 kg and being 3.1–5.7 years of age were used in the study. Prior to placement on study, NHPs were tested and verified seronegative by ELISAs for dengue viruses (DENVs), Japanese encephalitis virus (JEV), West Nile virus (WNV), and negative for ZIKV in both an ELISA and the plaque reduction neutralization test (PRNT). Further pre-screening requirements included confirmed negative for *Mycobacterium tuberculosis*, simian immunodeficiency virus (SIV), simian T-lymphotrophic virus-1 (STLV-1), simian retroviruses 1 and 2 (SRV-1 and SRV-2) via PCR, *Macacine herpesvirus* 1 (Herpes B virus), and *Trypanosoma cruzi* (ELISA and PCR). Following receipt of NHPs at Battelle, the NHPs were quarantined a minimum of 35 days prior to study start. Observations of all NHPs were performed a minimum of twice daily (at least 6 hours apart) during the quarantine period. NHPs were acclimated to pole/collar and chair restraint. NHPs were pair-housed in stainless steel cages to the extent possible, except for Study Days 56-70/71 when animals were housed individually, unless incompatible or for medical reasons as determined by a staff veterinarian. Cages were on racks equipped with automatic watering systems. The light/dark cycle was set to approximately 12 h each per day using fluorescent lighting. Animal room temperatures were controlled, monitored, and documented. The relative humidity of animal rooms was maintained. Certified Monkey Chow was provided and a Battelle veterinarian or other qualified individual reviewed the analysis report prior to use to ensure that no contaminants that would affect the results of the study were present in the feed. Water was supplied from the West Jefferson Municipal water system and was available *ad libitum* during the entire study per Battelle SOP. Water is analyzed at a minimum once per year. To promote and enhance the psychological well-being of NHPs, both food and environmental enrichment was provided to all NHPs.

The 22 rhesus macaques were randomized by animal ID to 1 of 3 test material groups of 5 animals each, and 1 'alum-only' control group of 5 animals. Specifically, males and females were allocated as follows: alum-only control group (3M,2F); 20μg VLP/300 μg alum (2M,3F);

5µg VLP/75µg alum (3M,2F); 1.25µg VLP/18.8 µg alum (2M,3F). The 2 extra animals (1 male and 1 female) were randomized as extras. The *in vivo* staff working with the animals were blinded. This included, test article administration, challenge, clinical observations, injection site observations, weights, and blood collections. In addition, the necropsy prosecutors and the histopathologist were also blinded to group assignment. Animals were randomized to a scheduled euthanasia day (Day 70 or 71) such that there were at least 2 animals per group on each scheduled euthanasia day. Blood samples were collected on the indicated days from a femoral artery or vein. At each hematology/clinical chemistry time point [Days 0, 7, 14, 28 (prior to VLP administration), 35, 56 (prior to ZIKV challenge), 62, 66, 70 or 71], approximately 1mL blood was collected from each animal into a tube containing EDTA and gently inverted 8–10 times to mix. At each serology time point, blood was collected from each animal into serum separator tubes (SSTs), gently inverted 4–8 times to facilitate contact of the blood with the separator to promote coagulation, allowed to clot, and processed for serum. On days with both clinical chemistry and immunogenicity analyses, 2 SST tubes were collected. SST tubes were spun at a relative centrifugal force (rcf) of at $1800 \times g$ for at least 10 minutes. Serum samples for immunogenicity analysis were stored at $\leq$ -70˚C until assessment or shipment. Telazol (2-3mg/kg, IM) or ketamine (3mg/kg, IM) was used for blood collections and rack changes as needed. Animals were anesthetized with Telazol (3mg/kg, IM) for test /control material administration and challenge. On Days 0 and 28, all animals received the test vaccine or control material via IM injection based on their randomized group assignment. We titrated both the VLP and alum doses in the NHP studies in contrast to the mouse studies that held the alum dose constant for each of the VLP titration doses. This was done to increase the likelihood of observing a vaccine 'dose-effect'. The volume of the vaccination material was 0.5mL per dose. The Day 0 administration occurred in the left thigh, whereas the Day 28 administration occurred in the right thigh. On Day 56, NHPs were challenged SC in the center of the back just caudal to the scapular region with a 1.0mL dose containing $4.74 \times 10^4$ PFU ZIKV as determined in residual challenge material that was stored on ice until assessment by plaque assay on Vero E6 cells.

## ZIKV-specific plaque and antibody assays

**Plaque assay.** A standard viral plaque assay was performed to determine viremia levels. At the time points, days 56 (prior to ZIKV challenge), 58–63, 66, 70 or 71 (days 0, 2–7, 10, and 14 or 15 post-challenge), blood samples were collected in SST tubes, processed to sera, and aliquots were made and stored at $\leq$ -70˚C. The assay LOD was 61 and LLOQ was 195 PFU/mL and titers less than LOD were assigned a titer of approximately one half the LOD (30 PFU/mL) for graphing and statistical purposes.

**Plaque reduction neutralization test (PRNT$_{50}$).** A plaque reduction neutralization test (PRNT) was also performed to evaluate ZIKV-neutralizing activity in serum samples collected at days 28 and 56 post-vaccination and at the post-ZIKV challenge timepoint of day 70 or 71. Blood samples were collected in SST tubes, processed to sera, and aliquots made and stored at $\leq$-70˚ C. Serial two-fold dilutions of heat-inactivated serum samples were incubated with an equal volume of well-characterized ZIKV strain PRVABC59 stock (BEI Resources, Manassas, VA) at 37˚C for 1 h. The serum-virus mixtures were then inoculated onto Vero E6 cells (BEI Resources) in 12-well tissue culture plates so that a target of 100 PFU ZIKV was inoculated into each well. After an incubation at 37˚C and 5% carbon dioxide for 1 h, a methylcellulose overlay medium was added to each well and plates were further incubated at 37˚C and 5% carbon dioxide for 72 h. Overlay medium was then removed, crystal violet stain solution (2.5% crystal violet, 15% formalin in distilled water) was added to fix and stain the cells, and plates

were incubated at room temperature for at least 20 min. Stain solution was then removed, wells were washed with distilled water, and plates were air-dried. Plaques were counted in each well and results were transcribed into a software analysis tool that uses a probit model to calculate the $PRNT_{50}$ titer, corresponding to the reciprocal of the serum dilution required to neutralize 50% of the input ZIKV infectivity using the average plaque count from wells inoculated with ZIKV only as a measure for 0% neutralization. The limit of detection for each titer was defined as the reciprocal of the lowest dilution evaluated in the assay, $PRNT_{50} = 10$. At the specified time points (Days 0 and 28 prior to vaccine administration, Day 56 prior to ZIKV challenge, and Day 70 or 71) blood samples were collected in SST tubes, processed to sera, and aliquots made and stored at $\leq$-70˚C. The ZIKV $PRNT_{50}$ was designed to quantify the ZIKV-neutralizing capability of Abs in serum using the fully characterized working bank of ZIKV strain PRVABC59 (Lot # 012017-ZIKVPRV-WS). NHP serum with no detectable ZIKV-neutralizing activity was used as the negative control serum, and NHP serum spiked with a ZIKV-neutralizing mAb was used as the positive control serum. In addition, the samples were tested both with and without the addition of 8% normal human serum (labile serum factor; LSF) to the Virus Diluent. Data reported in this manuscript were with the addition of 8% normal human serum. In absence of normal human serum, the data were generally of lower magnitude. Assay LOD was 10 and results below LOD were assigned a value of one half the LOD (5) for graphing and statistical analysis.

**Microneutralization ($MN_{50}$) assay.** A high-throughput microneutralization (MN) assay was performed to evaluate ZIKV-neutralizing activity in serum samples collected at days 28 and 56 post-vaccination and at the post-ZIKV challenge timepoint of day 70 or 71. This was a modification of an assay used to evaluate DENV neutralization responses in a dengue vaccine clinical trial [55]; the assay has been separately modified for evaluation of ZIKV-neutralizing activity in animal sera [56,57]. Serial three-fold dilutions of heat-inactivated serum samples were incubated with an equal volume of ZIKV strain PRVABC59 (stock originating from virus obtained from BEI) at 35˚C for 2 h. The serum-virus mixtures were then inoculated onto Vero E6 cells (BEI Resources) in 96-well microtiter plates so that a target of 100 plaque-forming units (PFU) ZIKV was inoculated into each well. These plates were incubated at 35˚C and 5% carbon dioxide for 72 h, fixed with cold 80% acetone for at least 30 minutes, fixative was removed, and plates were air-dried. The fixed, dried plates were rinsed three times with PBS and a horseradish peroxidase (HRP)-conjugated 6B6-C1 mAb (Hennessy Research; Shawnee, KS) was added to each well. Microtiter plates were incubated at 35˚C for 2 h, washed five times with PBS, and developed with 3,3´,5,5´-tetramethylbenzidine (TMB) substrate at 20–25˚C for 50 min. The colorimetric reaction was stopped using a 1:25 phosphoric acid in water and the optical density (OD) of each well was measured at 450 nm. From microtiter plates that met predefined system suitability criteria and on-plate control criteria (no-virus control average normalized OD $\leq$0.5; virus control average normalized OD $\geq$0.9), the median microneutralization ($MN_{50}$) for each sample was calculated using a log mid-point linear regression model and the $MN_{50}$ titer was reported as the reciprocal of the serum dilution that was required to neutralize 50% of the input ZIKV infectivity. The limit of detection was defined as the reciprocal of the lowest dilution evaluated in the assay, $MN_{50}$ titer = 10. NHP serum with no detectable ZIKV-neutralizing activity was used as the negative control serum and NHP serum spiked with a ZIKV-neutralizing mAb was used as the positive control serum. Assay LOD and assigning $MN_{50}$ values below LOD were the same as for $PRNT_{50}$.

**ZIKV *Renilla* luciferase neutralization assay (ZIKV-$RlucNT_{50}$).** ZIKV-specific nAb titer in serum samples collected at the designated time points was also determined using a luciferase-based virus neutralization assay. The assay is based on the capacity of Abs to neutralize recombinant Zika reporter virus, ZIKV-Rluc (Integral Molecular, Philadelphia, PA), with

readout in terms of the inhibition of luciferase transgene expression. Vero cells were propagated in DMEM containing 10% FBS and grown to 80–95% confluency in T225 flasks. The cells were harvested using trypsin-EDTA, seeded into 96-well tissue culture plates at $1.5 \times 10^4$ cells per well, and incubated overnight at 37±1˚C. Next day, serial 2.5-fold dilutions of test sera previously heat-inactivated at 56±1˚C for 45 min were prepared in dilution plates and incubated with a pre-determined fixed concentration of ZIKV-Rluc for 90 min. Final serum dilutions after the addition of virus ranged from 1:10 to 1:15,259. The serum virus mixtures were then added to the monolayers of Vero cells and plates incubated at 37˚C. Each test plate included a virus only control (VC), a positive ZIKV-specific Ab control, and a cell only control (CC). Following 48 h incubation, luciferase activity was measured using the Renilla Luciferase reagent system (Promega, Madison WI) and luminescence read on a SpectraMax M3 plate reader (Molecular Devices, Sunnyvale CA). Neutralization $NT_{50}$ titers were defined as the maximum serum dilution that neutralizes 50% of luciferase activity using linear regression analysis. Assay LOD and assigning $NT_{50}$ values below LOD were the same as for $PRNT_{50}$.

**Quantitative Reverse Transcriptase PCR, AG129 mice.**   At 5 days post-challenge, 200μL of whole blood in EDTA was collected via cheek bleed from AG129 mice in all groups. Serum fluid aliquot samples were frozen at −80˚C for RNA isolation and quantitative reverse transcriptase PCR (RT-qPCR). RNA was extracted using the Qiagen QIAamp cador Pathogen Mini Kit (Qiagen, Germantown, MD, USA) according to manufacture instructions and eluted with 100μL of elution buffer. RT-qPCR was performed as previously described [58] using a Mic-2 qPCR thermocycler (Bio Molecular Systems, Coomera, QLD, Australia). Standard curves of ZIKV RNA and *GAPDH* RNA were generated with serial dilutions of synthetic RNA (GeneScript, Piscataway, NJ, USA) of the target sequence (accession HQ234499.1). The relative number of ZIKV RNA genome equivalents (Relative GE) was determined by extrapolation from the ZIKV RNA standard curve and normalized to the relative *GAPDH* RNA level for each sample. Normalization to *GAPDH* was done to control against differences in RNA extraction. Serum samples with undetectable ZIKV RNA ($C_T \geq 40$) were arbitrarily assigned a $Log_{10}$ Relative GE value of -1 to visually differentiate them from ZIKV RNA positive samples (the lowest of which was -0.40) and for statistical analysis.

**Quantitative Reverse Transcriptase PCR, rhesus macaques.**   The ZIKV RT-qPCR was performed to evaluate viral loads following ZIKV challenge. RNA was isolated from serum samples collected on Study Days 0, 56, 58, 59, 60, 61, 62, 63, 66, and 70/71 (which are days -56, 0, 2–7, 10, and 14 or 15 post-challenge) using the cador Pathogen 96 QIAcube Kit (Qiagen, Strasse 1 40724 Hilden, Germany) on the QIAcube HT instrument (Qiagen) and stored at ≤-70˚C until RT-qPCR analysis. The isolated RNA was then evaluated for ZIKV genomic copies using primers and probe specific to the ZIKV prM-E gene region ([9]; Integrated DNA Technologies; Coralville, IA) and the RNA UltraSense One-Step Quantitative RT-PCR System (Invitrogen; Carlsbad, CA) on a QuantStudio Flex 6 Real-Time PCR System (Applied Biosystems; Foster City, CA). A standard curve for absolute quantitation of ZIKV genome copies was included on each assay plate using a synthetic RNA (Bio-Synthesis, Inc.; Lewisville, TX) containing the amplicon sequence targeted within the PrM-E gene region of ZIKV strain PRVABC59 (GenBank Accession Number KX087101.3). Thermocycling times were: Stage 1–50˚C for 15 min for one cycle; Stage 2–95˚C for 2 min for one cycle; Stage 3–95˚C for 15 sec and 60˚C for 1 min for 40 cycles. Data analyses were performed using the QuantStudio 6 software-generated values (total genome copies per well of each sample) and additional calculations to determine ZIKV genomic copies per mL of serum. The RT-qPCR assay had no established LOD or LLOQ for NHP serum so all values of genome equivalents per mL serum (GE/mL) are presented and an arbitrary value of $10^0$ GE/mL was assigned to serum samples with undetectable ZIKV RNA for graphing and statistical analysis.

## Statistical analysis

GraphPad Prism 8.3.0 was used to plot results and for statistical analyses. The statistical analyses used for each dataset are also described in each Fig caption. Where noted, assay results below the detection limit were assigned a value one half the detection limit for graphing and statistical analysis. Serum nAb titers generated by $PRNT_{50}$, $MN_{50}$, and $RlucNT_{50}$ assays; infectious virus titers by plaque assay; and RNAemia levels by RT-qPCR (Log Relative GE for mice and GE/mL for NHP) were compared within each time point by Kruskal-Wallis nonparametric ANOVA followed by Dunn's comparison tests between all pairwise comparisons. The significance values compared to the alum only group are shown in each Fig. Survival rates relative to the alum only group were compared using Fisher's exact test. Percentage weight changes in mice post-ZIKV challenge were compared by one-way ANOVA followed by Dunnett's multiple comparison tests with the uninfected control group. Day 49 Log Relative GE values and Day 42 $RlucNT_{50}$ levels in the surviving versus dead mice were compared by Mann-Whitney U tests. Regression analyses of 1) Day 42 $RlucNT_{50}$ titers compared to Day 49 Relative RNA GE values (mouse), 2) Day 56 $PRNT_{50}$ titers compared to $MN_{50}$ titers, 3) Day 56 $PRNT_{50}$ titers compared to $RlucNT_{50}$ titers, 4) Day 56 $PRNT_{50}$ titers compared to Day 59 GE/mL levels, and 5) $RlucNT_{50}$ vs Day 59 GE/mL titers were performed on $Log_{10}$-transformed values by least squares regression and R values and P values are shown in each Fig. All P values are 2-tailed and significance levels are denoted as: ****$P < 0.0001$, ***$P < 0.001$, **$P < 0.01$, *$P < 0.05$, or ns (not significant). The complete set of raw data is available as Supporting Information (S1 Data).

## Results

### ZIKV VLP plasmid construction

The ZIKV VLP hybrid is comprised of the prM-E structural proteins derived from the MR766 African strain with the exception that the E ectodomain comes from the SPH2015 Brazilian strain (Fig 1A). The rationale for this construct was based on selecting a VLP vaccine candidate that presented a structure that induces a nAb response specific for a recent circulating Brazilian strain [59,60].

### VLP production, purification, and characterization

HEK293 cells were transfected with the ZIKV VLP plasmid construct using electroporation. Four days following transfection, cell supernatant was harvested and clarified by centrifugation. Column chromatography was used to purify the VLPs from host and media contaminants. VLPs were evaluated post-purification by SDS-PAGE with Coomassie Blue staining (Fig 1B) and Western Blotting (Fig 1C). The Coomassie gel (lanes 1 and 2), indicates a highly purified VLP protein preparation that, when transferred to a Western Blot (lanes 1 and 2), shows a 55kDa band that was detected using an E-specific Ab.

### ZIKV VLP immunogenicity and efficacy in AG129 mice

The immunogenicity of the ZIKV VLP vaccine candidate was tested in AG129 mice at doses of 0.1, 1.0, and 10µg with and without an alum adjuvant (100µg dose) and administered via IM injection (Fig 2A). Groups of 10 adult male and female AG129 mice were vaccinated with ZIKV VLP following a prime/boost schedule prior to challenge on Day 44 with a Puerto Rican isolate of ZIKV, PRVABC59 at a dose of ~100 $CCID_{50}$ delivered by SC injection into the inguinal space between the back-right leg and the body in 0.1mL volume. Blood draws were taken following each immunization on Days 14 and 42 for determination of virus nAb responses

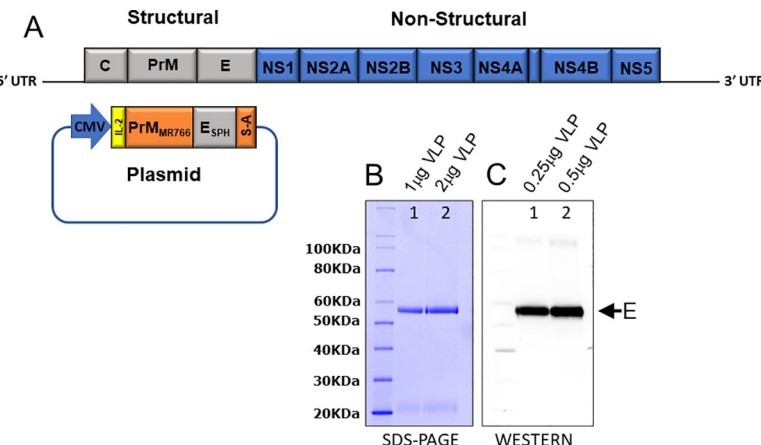

**Fig 1. Design, production, purification, and characterization of ZIKV VLP vaccine candidate.** (1A) Schematic of ZIKV genome and plasmid DNA encoding ZIKV structural genes: 1) prM derived from the MR766 African strain (orange) with native signal sequence replaced with the human IL2 sequence (yellow) (MYRMQLLSCIALSLALVTNS); 2) E ectodomain derived from the SPH2015 Brazilian strain (grey), and 3) E stem-anchor (S-A) from MR766 (orange). (1B) SDS PAGE Coomassie gel, lane 1, 1µg purified VLP, and lane 2, 2µg purified VLP, approximately 55kDa. (1C) Western blot, lane 1, 0.25µg purified VLP, and lane 2, 0.5µg purified VLP, approximately 55kDa.

prior to viral challenge. Significant nAb responses, as measured by the $RLucNT_{50}$ assay, were evident for the 10µg VLP +/- alum and 1µg VLP + alum dose groups relative to the alum only group following the prime immunization (Fig 2B). The prime nAb responses were increased from 10 to 80-fold following the second, booster immunization (Fig 2C). The increase was

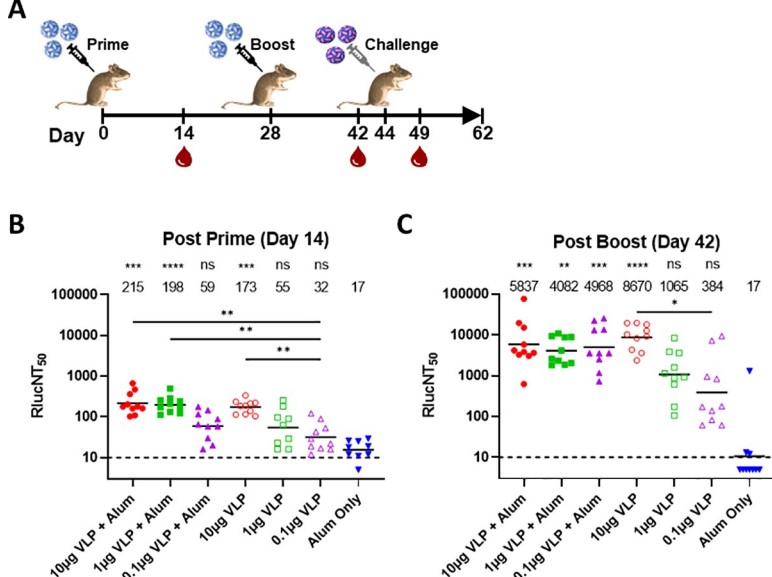

**Fig 2. Zika VLP mouse study timeline and serum nAb responses.** (2A) Schematic of experiment design. AG129 mice immunized either with Zika VLP or alum only (100µg) as control (n = 10). (2B) Serum $RlucNT_{50}$ titers post prime on Study Day 14 and (2C) post boost on Study Day 42 were determined. GMTs for each group are shown by lines and values are given above each group. $NT_{50}$ titers of individual mice are shown by symbols. The assay LOD was 10 (dashed lines) and titers below LOD are shown as 5. Kruskal-Wallis analysis followed by Dunn's comparison tests on all pairwise comparisons within each time point were performed and statistical significance levels relative to each alum only group are denoted above the GMT value as: $^{****}P < 0.0001$, $^{***}P < 0.001$, $^{**}P < 0.01$, $^{*}P < 0.05$, or ns (not significant). Additional significant differences between pairwise VLP groups are shown by bars and significance levels.

most evident at the lowest dose of 0.1μg formulated with alum where the geometric mean (geomean) nAb titer was boosted approximately 80-fold, 59 to 4,968. Vaccination, in the absence of adjuvant, following a booster immunization was dose-responsive, with geomean nAb titers of 384 (0.1μg), 1,065 (1.0μg), and 8,670 (10μg). VLP vaccine formulated with alum adjuvant resulted in consistent production of geomean nAb titers in the range of 4,000–6,000 regardless of dose, Fig 2C. Of note, alum adjuvant provided dose-sparing effect that was most evident at the lowest, 0.1μg dose. Although not a significant difference, following the booster immunization, the geomean nAb titer was increased approximately 13-fold when the VLP was formulated with alum, i.e., 384 to 4,968, Fig 2C.

VLP vaccine efficacy was evaluated in the AG129 lethal ZIKV challenge mouse model, Fig 3. Mice were immunized with a dose titration (10, 1.0, 0.1μg dose formulated +/- 100μg alum) of ZIKV VLP (50μL each hind limb, 100μL total) by the IM route. Mice were challenged on Day 44 with ZIKV strain (PRVABC59) with ~100 CCID$_{50}$ delivered by SC injection into the inguinal space between the back-right leg and the body in 0.1 mL volume. Complete protection was observed in mice vaccinated with the highest 2 doses of alum adjuvanted ZIKV VLP, with 90% protection at the lowest dose, 0.1μg VLP + alum (Fig 3A). Mice vaccinated with ZIKV VLP alone resulted in 100%, 70% and 50% survival in mice vaccinated with 10, 1 or 0.1μg of VLP, respectively. Significant differences in protection of vaccinated mice compared to alum only administered mice are indicated. In addition, the 0.1μg VLP group survival was significantly lower than that of the 3 fully protected groups (10μg VLP + Alum, 1μg VLP + Alum, and 10μg VLP, P = 0.033 Two-tailed Fishers Exact Test), but no other statistically significant differences were achieved. Shown in Fig 3B are the average weight changes for mice challenged with ZIKV. On Days 9, 11, 13, and 15 post-challenge, mice immunized with VLP with or without alum adjuvant did not have a significant weight change compared to uninfected control mice. However, the alum only administered group had a significantly different weight change compared to the uninfected controls on the indicated days.

Relative viral Genome Equivalents to GAPDH RNA (Relative GE) in Day 5 post-challenge serum were significantly reduced in a dose-responsive manner in all vaccinated mice, (except 0.1μg VLP group) as compared with infected (alum only) control (Fig 4A). The alum only control mice exhibited 100,000 geomean ZIKV RNA copies per GAPDH copy and succumbed to infection. Mice that did not survive are indicated by a '†' symbol. VLP immunized mice that were protected had significantly decreased serum geomean relative RNA copy levels compared

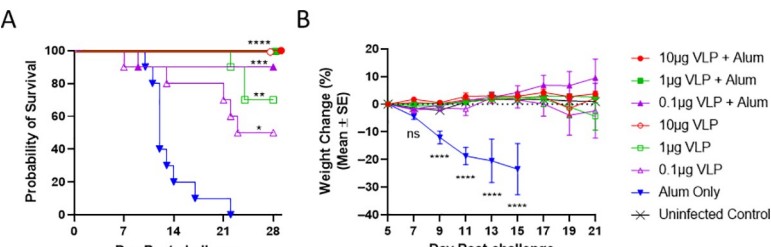

**Fig 3. Survival and weight loss following lethal ZIKV challenge.** On day 44 of the study, mice were given an SC challenge of ~100 CCID$_{50}$ of ZIKV strain PRVABC59. (3A) Survival was monitored for 28 days and survival rates for each VLP group were compared to the alum only (100μg) group by Fisher's exact test with significance levels denoted as: ****P < 0.0001, ***P < 0.001, **P < 0.01, *P < 0.05, or ns (not significant). (3B) Weight loss was monitored for 16 days starting at day 5 post-challenge. Group mean weight loss is shown as a percentage relative to day 5 post-challenge with error bars denoting group standard error of the mean for ease in viewing. An uninfected control group (n = 5) is shown for comparison and the dotted line indicates zero change. Weight changes were compared on days 7 through 15 by one-way ANOVA, and comparisons to the alum only group were by Dunnett's multiple comparison tests with significance levels denoted as for Fig 3A.

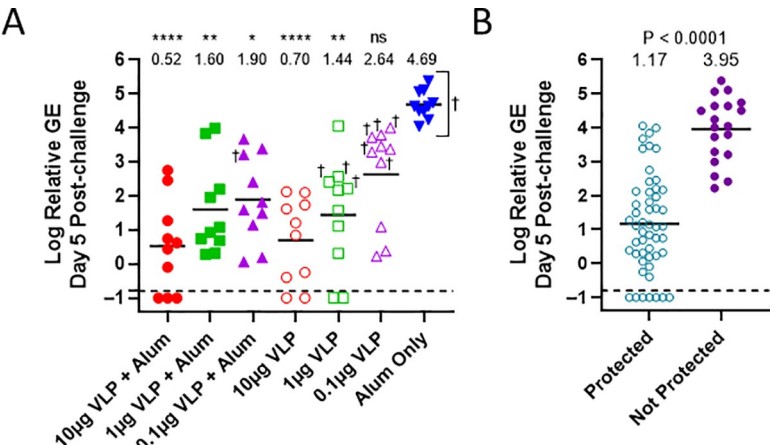

**Fig 4. Protection against ZIKV RNAemia following challenge.** (4A) Serum was collected on day 5 post-challenge (Study Day 49) for quantification of ZIKV genomes by RT-qPCR. Each sample was analyzed against ZIKV RNA and GAPDH RNA standard curves and the ZIKV genome equivalents (GE) relative to GAPDH were determined and $Log_{10}$ transformed. Serum samples with no detectable ZIKV RNA were assigned a $Log_{10}$ Relative GE value of -1 for graphing and statistical analysis. Group mean Relative GE levels are shown by line, individual mice shown by symbol, and '†' denotes the individual mice that succumbed to infection. All mice in the alum only (100µg) control group died. The dashed line indicates the highest Log relative GE from the sera of 5 naïve mice (-0.77) and is shown for comparison. Statistical significance relative to the alum only group by Kruskal-Wallis test followed by Dunn's pairwise comparison tests are denoted as: ****$P < 0.0001$, ***$P < 0.001$, **$P < 0.01$, *$P < 0.05$, or ns (not significant). All possible pairwise comparisons were performed and there were no statistically significant differences among VLP groups. (4B) $Log_{10}$ relative ZIKV GE on Study Day 49 (day 5 post-challenge) were plotted compared to mice surviving (protected) and mice succumbing to infection (Not Protected). Group mean $Log_{10}$ Relative GE are denoted by line, individual titers by symbols, the maximum naïve mouse value by a dashed line as in Fig 4A, and P value was calculated by Mann-Whitney U test.

to those succumbing to infection; 15 compared to 8,913 copies, respectively, Fig 4B. There were no statistically significant differences in the GE levels between VLP groups.

We next sought to address whether there was a correlation between the level of nAb achieved following VLP immunization and protection, Fig 5. nAb levels following the second VLP immunization were plotted to compare individual mice that were protected (survived) or not protected (succumbed to infection), Fig 5A. There was a significant difference between

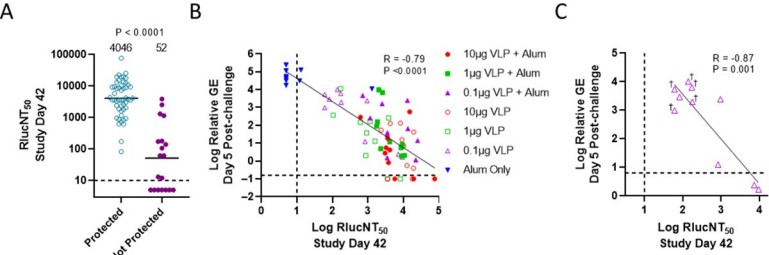

**Fig 5. Neutralizing antibody levels induced by ZIKV VLP vaccination correlates with protection.** (5A) Comparison of Day 42 $RlucNT_{50}$ titers in mice that survived (Protected) or died (Not Protected) after lethal ZIKV challenge. GMTs is shown by bars, individual titers by symbols, and assay LOD by dashed line. Significance value shown is from Mann-Whitney U test. (5B) Correlation of Day 49 (day 5 post-challenge) relative ZIKV GE compared to Day 42 $RlucNT_{50}$ titers for all groups. (5C) Correlation of Day 49 (day 5 post-challenge) relative ZIKV GE compared to Day 42 $RlucNT_{50}$ titers for the VLP 0.1µg dose group only. Log-transformed values for each serum $RlucNT_{50}$ titer and Relative GE RNAemia load (symbols based on immunization group), with dashed lines representing the assay LOD for $RlucNT_{50}$ assay (x-axis) or reference level for maximum naïve mouse Relative ZIKV GE as in Fig 4 (y-axis) are provided. R and P values as well as regression lines shown were determined by least squares regression analyses.

groups with a geomean nAb titer of 4,046 (protected) compared to 52 (not protected). Also, the correlation between Day 49 (day 5 post-challenge) Relative RNA copy level compared to Day 42 nAb level was evaluated first using all of the mice in the study, Fig 5B. The variables were inversely correlated (R = - 0.79, P = < 0.0001) indicating that induction of nAb may be needed to afford protection. Next, the correlation of nAb and Relative RNA copy level was examined for the 0.1μg VLP group, Fig 5C. Of note, the VLP group that had the highest post-challenge RNA load (Fig 4A) and lowest survival rate (Fig 3A), i.e., the 0.1μg VLP alone group, showed a strong significant inverse correlation (Fig 5C) (R = -0.87, P = 0.001). The 2 other VLP immunized groups that had suboptimal protection showed significant correlations between nAb and RNA copy levels, i.e., the 1μg VLP + alum (R = -0.691, P = 0.027) and 1μg VLP (R = -0.729, P = 0.0168) groups. This study demonstrated potent protective efficacy of a ZIKV VLP vaccine in a lethal mouse model that was correlated with level of nAb induced.

## ZIKV VLP immunogenicity and efficacy in rhesus macaques

The VLP vaccine was next evaluated in an NHP model. Rhesus macaques (5 per group) were immunized on Days 0 and 28 with various dose levels of VLPs formulated with alum (20μg VLP/300μg alum; 5μg VLP/75μg alum; and 1.25μg VLP/18.8μg alum) followed by a ZIKV PRVABC59 challenge on Day 56, Fig 6. Challenge virus was administered SC in the center of the back just caudal to the scapular region with a 1.0mL dose containing $4.74 \times 10^4$ PFU virus. Blood samples were drawn following each VLP immunization (Days 28 and 56), and viral

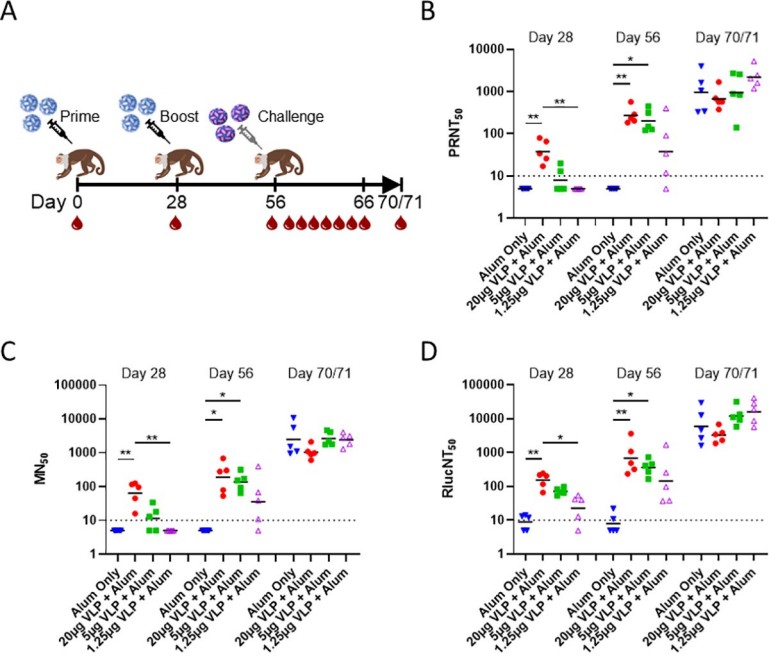

**Fig 6. Experimental design and serum neutralizing antibody responses to Zika VLP vaccination in NHPs.** (6A) Rhesus macaques (5 per group) were immunized on the indicated days with a dose titration of VLPs formulated with alum (20μg VLP/300μg alum; 5μg VLP/75μg alum; and 1.25μg VLP/18.8μg alum) or alum only (300μg) (black syringes), test bled (red drops), and challenged (gray syringe) on the days shown. (6B) Serum nAbs measured by $PRNT_{50}$, (6C) microneutralization ($MN_{50}$) assay, and (6D) *Renilla* luciferase (Rluc)-based ZIKV neutralization assay ($RlucNT_{50}$). Test bleed time points are shown at top for each assay. GMTs are shown by bars, individual titers by symbols, and assay detection limits by dotted lines. Levels of significance between groups are shown by lines and were obtained by performing Kruskal-Wallis and Dunn's pairwise comparison tests denoted by: ****P < 0.0001, ***P < 0.001, **P < 0.01, *P < 0.05, or ns (not significant).

challenge (Days 56, 58–63, 66, and 70/71 of the study which are days 0, 2–7, 10, and 14/15 post-challenge), Fig 6A. ZIKV VLP-induced humoral immunity was evaluated on Days 28, 56, and 70/71 by virus-specific nAb assays: PRNT, MN, and RlucNT. Protection following viral challenge was evaluated by viremia (PFU/mL) and viral RNA genome equivalents (GE/mL) observed in the plasma on Days 56, 58–63, 66, and 70/71 which are days 0, 2–7, 10, and 14/15 post-challenge.

Vaccine safety was also evaluated whereby extensive clinical observations, hematology, and clinical chemistry changes were recorded for the rhesus macaques. This was not the focus of this study. However, it should be noted that no local abnormal injection site observations, nor systemic clinical or hematological changes (besides minor stool abnormalities) were observed following vaccination or viral challenge. In addition, all animals gained weight from arrival through end of the in-life study.

ZIKV-specific nAbs were generated in a dose dependent manner in animals receiving ZIKV VLP as measured by $PRNT_{50}$, $MN_{50}$ and $RlucNT_{50}$ assays through day 70/71 (Fig 6B–6D, respectively). By Day 28, following the primary immunization, only the high dose group of ZIKV VLP 20μg induced significant nAb responses compared to the alum only group (300μg) as evaluated by $PRNT_{50}$, $MN_{50}$, and $RLucNT_{50}$, with geomean titers for the 20μg group being 38, 63, and 153, respectively. On Day 28, the 20μg VLP dose induced nAb titers that were also significantly higher than those induced by the 1.25μg dose. There were no significant differences between other pairwise VLP dose comparisons. Following the booster immunization (Day 56), both the VLP 20 and 5μg doses induced significant nAb responses compared to the alum only group. However, there were no significant differences in nAb responses between the different VLP dose groups. The geomean nAb responses induced by the VLP 20μg dose were 270 ($PRNT_{50}$), 189 ($MN_{50}$) and 679 ($RlucNT_{50}$) while the nAb responses observed for the 5μg dose were 200 ($PRNT_{50}$), 137 ($MN_{50}$) and 360 ($RlucNT_{50}$). At study end, Day 70/71, geomean nAb responses were elevated compared to the Day 56 viral challenge time point, Fig 6B–6D. At this final time point there were no significant differences in responses between VLP dose groups or between dose groups and the alum only group. The end-of-study responses, in part, likely indicate de novo synthesis of nAb production based on viral replication.

In order to better understand how the $MN_{50}$ and $RlucNT_{50}$ assays compared to the gold standard $PRNT_{50}$ assay, we examined the correlation strengths between assay results, Fig 7. The $MN_{50}$ and $PRNT_{50}$ nAb titers were highly correlated (R = 0.84, P <0.001) as shown in Fig 7A. A statistically significant correlation was also observed, albeit at a lesser extent, when

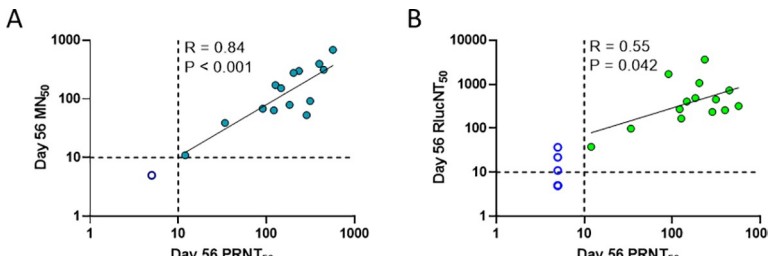

**Fig 7. Correlation between serum neutralization titers determined by $MN_{50}$ or $RlucNT_{50}$ and $PRNT_{50}$.** (7A) Correlations were evaluated for $MN_{50}$ compared to $PRNT_{50}$ and for (7B) $RlucNT_{50}$ compared to $PRNT_{50}$ day 56 serum samples, with resultant R and P values shown and non-Log transformed values plotted. Closed circles depict individual monkey serum samples that had detectable titers for both assays and that were included in the regression analysis, while open circles depict sera that were negative for at least one assay and excluded from analysis. R and P values and regression lines shown by least squares regression. Dashed lines show assay detection limits. Any titers below the limit of detection were assigned a value of 5 for graphing and visualization purposes.

comparing $PRNT_{50}$ and $RlucNT_{50}$ (R = 0.55, P = 0.04), Fig 7B. It should be noted that the $RlucNT_{50}$ assay has a higher detection frequency for low-level nAb titers and higher dynamic range (approximately 3 Log range compared to 2) which may, in part. explain why $RlucNT_{50}$ has the lower correlation. The rationale for developing the $RlucNT_{50}$ assay is due to it being more amenable to high throughput which is especially needed when evaluating large numbers of serum samples obtained in clinical trial studies.

The ability of the ZIKV VLP vaccine to prevent viremia as monitored by a plaque assay was evaluated on Days 0, 2–7, 10, and 14/15 post-challenge (Fig 8). Peak viremia occurred on days 2 and 3 post-challenge and no viral plaques were observed by 5 days post-challenge (Fig 8A). Only 3 control animals in the alum only group and 1 animal in the low dose vaccine group (VLP 1.25µg) were positive for viremia in at least 1 plaque assay. Specifically, the following NHPs in the alum only group were observed with viremia: NHP #RA3056 (133 plaques on day 3 post-challenge); NHP #RA3111 (133 and 100 plaques on days 2 and 3 post-challenge, respectively); and #RA3281 (944 and 278 plaques on Days 2 and 3 post-challenge, respectively). NHP #RA2555 in the VLP 1.25µg + alum dose group demonstrated viremia, i.e., 100 viral plaques on day 4 post-challenge. No statistically significant differences in viral load on any day were observed (Kruskal-Wallis plus all possible Dunn's pairwise comparison tests). Testing for infection by RT-qPCR was also performed, and viral replication, as measured in serum as RNA copies (Genome Equivalents/mL), was detected in all groups except the VLP 5µg dose group (Fig 8B). All 5 animals in the control group (alum only) were positive with viral RNA copies measured in the range of $1.9 \times 10^2$ to $4.2 \times 10^6$ over days 2 to 6 post-challenge. No detectable RNA copies were observed after day 6 post-challenge in any of the groups (Fig 8B). There was no detection of RNA copies in the 5µg VLP intermediate dose group with significant differences (by Kruskal-Wallis analysis plus Dunn's multiple comparison tests on all pairwise comparisons) in RNA copies compared to alum only control group noted: days 2 (P = 0.0049); 3 (P = 0.0051); 4 (P = 0.0041); and 5 (P = 0.0369) post-challenge. Whereas, in the 20µg VLP high dose group, low RNA copies of 99 and 194 for 2 individual animals for only a single day each were observed with significant differences in RNA copies again noted compared to the alum only control group: days 2 (P = 0.0049); 3 (P = 0.0171); 4 (P = 0.0123); and 5 (P = 0.0369) post-challenge. Breakthrough of RNA copies was observed in 4 of 5 animals in the 1.25µg VLP low dose group with RNA copies measured in the range of $3.2 \times 10^3$ to $1 \times 10^6$ on day 3 and $1.3 \times 10^2$ to $2.2 \times 10^5$ on day 4 post-challenge. There were no statistically significant differences in RNA copies between 1) the 1.25µg VLP dose group and alum only control group over days 2 through 5 post-challenge or 2) any of the VLP groups with each other on any day except for the 5µg VLP dose group.

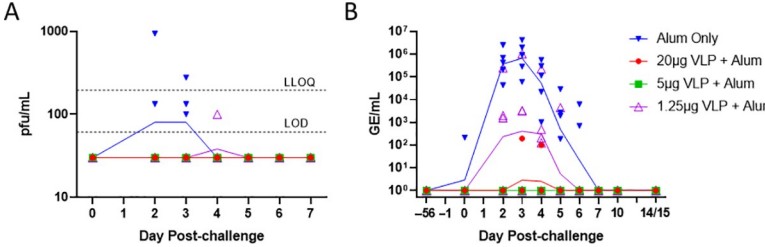

**Fig 8. Protection against ZIKV viremia following challenge.** Following ZIKV challenge on day 56, blood samples were tested for infectious virus by (8A) plaque assay and (8B) RNAemia by RT-qPCR. GMT for each titer is shown by a line, individual titers by symbols, and plaque assay LLOQ (195 PFU/mL) and LOD (61 PFU/mL) by dashed lines. Note that the RT-qPCR assay does not have an established LLOQ or LOD, so all data are presented and samples with undetectable ZIKV RNA shown at $10^0$ GE/mL. Statistical differences between groups (by Kruskal-Wallis analysis plus Dunn's multiple comparison tests for all pairwise comparisons) are indicated in the text.

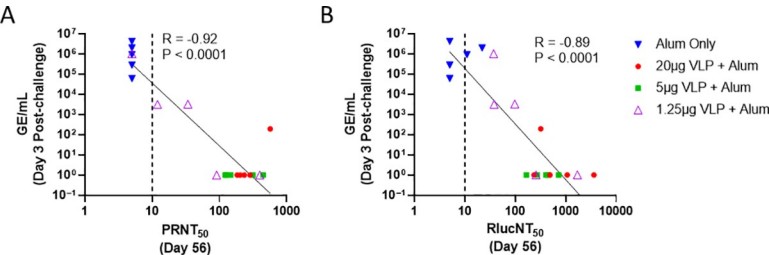

**Fig 9. Correlate of protection by ZIKV VLP vaccine.** Day 3 post-challenge ZIKV GE/mL (RNAemia) compared to day 56 nAb titers obtained by (9A) $PRNT_{50}$ and (9B) $RlucNT_{50}$ assays are plotted for each animal with immunization groups denoted by symbol. R and P values and regression lines are shown and non-Log transformed values plotted. The dashed line shows the LOD for the $RlucNT_{50}$ assay, and RNA GE/mL levels that were undetectable were assigned a value of $10^0$.

The correlation between nAbs levels elicited by VLP immunization and subsequent levels of ZIKV RNAemia following challenge was examined, and significant inverse correlations were observed on day 3 post-challenge between GE/mL and $PRNT_{50}$ (R = -0.92, P < 0.001) (Fig 9A) and between GE/mL and $RlucNT_{50}$ (R = -0.89, P < 0.0001) (Fig 9B) supporting nAb as a potential correlate of protection. The $RlucNT_{50}$ assay results were comparable to $PRNT_{50}$ results with respect to correlating with protection against RNAemia. Taken together, these results indicate that ZIKV VLP induced ZIKV-specific nAbs in a dose-dependent manner and protected against ZIKV infection as measured by viral RNA replication (RT-qPCR) assays.

## Discussion

The advantages of a using a VLP platform for vaccine development include: 1) intrinsic safety of non-replicating subunit vaccines, 2) no likelihood of reversion or recombination events as in live-attenuated vaccines, 3) no need for chemical or other harsh inactivation steps in the case of inactivated vaccines, and 4) production of particles that accurately reflect viral structure in regard to induction of viral epitope-specific Ab and cellular immune responses [61]. Of clinical relevance, several VLP-based vaccines are approved for human use: human papilloma virus, hepatitis B virus, hepatitis E virus, and malaria [62].

In the studies described herein, we evaluated a VLP-based approach for ZIKV vaccine development. Transient transfection via electroporation of mammalian HEK293 cells, using a plasmid construct encoding the ZIKV prM-E structural genes, resulted in production of VLPs which were readily purified from culture supernatants using column chromatography. VLPs formulated with an alum adjuvant and administered to AG129 mice and rhesus macaques generated high-titer nAb responses. Correlate of protection analysis supported a correlation between the level of nAb induced by VLP vaccination and the level of protection as measured by survival (mice) and viral replication (mice, NHPs). An earlier study by our group demonstrated that passive transfer of mouse immune sera induced by a ZIKV VLP immunization protected AG129 mice against a lethal ZIKV challenge, demonstrating the importance and sufficiency of nAb responses in protection against ZIKV by the VLP vaccine [36].

Here, we sought to extend our VLP vaccine development studies to include the rhesus macaque model. Rhesus macaques have been used as models of ZIKV disease and also as vaccine efficacy [56,63–67]. In regard to our NHP ZIKV challenge studies, it was demonstrated that only 3 of 5 alum only immunized control NHP animals demonstrated infection detectable by plaque assay. This observation suggested that a higher challenge dose of ZIKV or a different species of NHP may have provided a more robust model in regard to evaluation of protection by viremia. It should be noted that at low titer passage of PRVABC59 has been shown to

acquire mutations in E (V333L) and NS1 (W98G) which attenuates pathogenesis in mice [68]. It is not known how these mutations affect viral pathogenesis in NHPs. The virus was sequenced prior to the NHP challenge whereby the consensus sequence (more than 50% of the sequences assessed) did not have the E (V333L) mutation but did have the NS1 (W98G) mutation in approximately 62% of the sequences tested. This mutation may or may not be a contributing factor in low level of viremia detected in the NHPs. In contrast to viremia, efficacy was clearly shown by the detection of ZIKV RNA in all 5 animals in the alum only group using the more sensitive RT-qPCR assay and statistically significant RNA copy reductions in the 2 higher dose VLP vaccinated groups. Other groups typically use RNAemia as measured by RT-qPCR as a readout of efficacy. Viral challenge doses as low as $1 \times 10^3$ PFU or FFU achieved levels of $1 \times 10^{4-7}$ copies of RNA/mL which is comparable to levels noted in our studies [56,69]. Cynomolgus monkeys have also been used as a model for RNAemia demonstrating again comparable levels of viral RNA copies in the blood following viral challenge in the range of $1 \times 10^{4-5}$ [70,71].

Other groups have utilized the prM-E structural genes as a ZIKV vaccine approach using delivery formats other than our VLP candidate vaccine, e.g: DNA ([29]; Ph1 NCT02809443; Ph1 NCT02887482; Ph2 NCT02996461); RNA ([72]; Ph1 NCT03014089), inactivated virion ([33]; NCT02963909; NCT02952833; NCTO2937233; CTRI/2017/05/008539; NCT03343626; NCT03425149); or viral-vectored [35], e.g., dengue, vaccinia, adenovirus, and measles (Ph1 NCT02996890). Differences in VLP composition were made by addition of the capsid protein to the prM-E structural proteins [73,74]. Garg and colleagues provided data suggesting that superior nAb titers were induced by VLPs expressing the capsid protein compared to VLPs expressing only prM-E, and this effect was observed at the lowest dose tested, 3μg. Additional animal studies and even immunogenicity comparisons in humans would likely be required to confirm that capsid inclusion in the VLP structure results in a more potent immunogen. In summary, differences in VLP vaccine development are based on delivery platform utilized (encoded in nucleic acid or delivered as a particle), composition of VLP (prM-E +/- capsid structural proteins), and ZIKV strain sequences used in VLP composition. Of note, the following are examples of ZIKV strains used in VLP composition; 1) the NIAID/VRC DNA vaccine encodes sequences from the French Polynesian isolate strain H/PF/2013, [69]; the Moderna mRNA vaccine encodes sequences from the Asian ZIKV strain Micronesia 2007 [72], and as noted previously, our VLP E ectodomain is derived from the Brazilian strain SPH2015. The rationale for strain selection, in part, is based on belief that immune responses specific for these ZIKV structural genes will be protective against current circulating strains.

Adjuvants are often needed for non-replicating vaccines to increase immunogenicity and efficacy. The mouse studies provided herein suggest that alum had a dose-sparing effect. This trend was most evident at the lowest, 0.1μg dose, Fig 2. Following the booster immunization, the geomean nAb titer was increased approximately 13-fold. Aluminum salts are effective adjuvants that have been used in human vaccines for almost a century. A large amount of data collected shows that aluminum adjuvants with various vaccine antigens have highly effective capability to enhance antibody responses, are generally well tolerated, do not cause pyrexia, and have the strongest safety record of any human adjuvant [75]. Furthermore, the dose of aluminum (300μg) used in the NHP study is consistent with the general biological product standards and are well within the limits referenced in 21CFR610.15. As a precedent, the aluminum hydroxide adjuvant is contained in numerous U.S.-licensed vaccines (GARDASIL 9, Infanrix, Kinrix, Pediarix, Havrix, Engerix-B, Ixiaro, Bexsero, and Boostrix).

An important consideration, for this and other ZIKV vaccines, in addition to vaccination of an at-risk population to establish herd immunity, is whether VLP immunization induces the quality and magnitude of an immune response that would be capable of protecting a

developing fetus. In this report, we demonstrated in the AG129 mouse and NHP animal models that prophylactic administration of VLP could protect against death (mice) and limit viral replication (mouse and NHPs). However, the nAb titers increased following viral challenge in NHPs, indicating lack of sterilizing immunity. As noted in Fig 6 following viral challenge, the geomean nAb titer increase at Day 70/71 was inversely proportional to the immunizing dose. For example, at the 20μg VLP dose, Day 56 RlucNT$_{50}$ nAb titer of 679 increased to 3,303 by Day 70/71 (5-fold increase) vs. the 1.25μg VLP dose at Day 56 of 143 titer increased to 15,996 (112-fold increase) by day 70/71. There was a trend, although not significant, that the lowest VLP 1.25μg dose resulted in a higher geomean nAb titer compared to the alum only group following viral challenge, potentially suggesting B cell restimulation by breakthrough viral replication or Ab-dependent enhancement of virus infection after challenge. Animal models [66,67,76,77] may be of benefit to evaluate whether immune responses generated by the VLP are sufficient to prevent fetal harm. Van Rompay and colleagues evaluated the VRC's ZIKV prM-E DNA vaccine in a ZIKV-exposed pregnant macaque model. Vaccine-induced nAb titers were positively associated with reduced maternal viremia and subsequent protection of the fetus [66]. Field efficacy trials will likely be needed to truly answer this question [78].

We compared our preclinical immunogenicity and efficacy results to those of other ZIKV vaccine platforms. In general, our VLP vaccine induced nAb responses that provided protection levels against ZIKV challenge that were comparable or superior to the other vaccines. However, it should be noted that it is difficult to directly compare the nAb assays between laboratories. Gaudinski and colleagues reported on a ZIKV DNA vaccine encoding prM-E in Ph 1 studies [29]. Earlier studies evaluating this vaccine in rhesus macaques demonstrated that a nAb titer of 1:1,000, analyzed using ZIKV reporter virus particles, protected 70% of the animals against viremia following ZIKV challenge [69].

In preclinical studies, an RNA vaccine against ZIKV was evaluated in rhesus macaques [65]. Macaques were immunized by the intradermal route with 50, 200, or 600μg of nucleoside-modified prM-E lipid nanoparticle encapsulated mRNA which induced FRNT$_{50}$ (format similar to PRNT$_{50}$) nAb titers of approximately 400 which upon viral challenge completely protected macaques against detection of RNA copies in the plasma.

Efforts to develop inactivated ZIKV vaccines are also currently ongoing. Modjarrad and colleagues evaluated an inactivated ZIKV vaccine in Ph 1 studies ([30]; NCT02963909; NCT02952833; and NCT02937233). The vaccine (5μg given with AlOH adjuvant) was administered twice in 4 weeks and importantly, 92% of the vaccine recipients seroconverted and reached the primary immunogenicity endpoint of GMT at day 57 of 1:100. This level exceeded the thresholds for protection in non-human primate studies where MN$_{50}$ nAb titers of 1:60 protected against detection of ZIKV RNA copies in the plasma following viral challenge [56].

With the current widespread endemic transmission of ZIKV and the potential for further epidemic spread, an effective vaccine is still needed. The timing and severity of the next outbreak is unpredictable, but seems inevitable. Potential outbreaks in naïve populations always pose a danger with outbreaks likely delayed for regions that already experienced epidemics. Counotte and colleagues evaluated future age-specific risk using data from Managua, Nicaragua [79]. They assumed lifelong immunity and predicted that threat of an outbreak will remain low in the region until 2035 and increase to 50% in 2047 in the 15–29 age group where pregnancies will be at the highest risk of infection. The risk of infection will likely increase if there is not lifelong immunity and/or low levels of nAb enhance infection. For these reasons, the continuing development of a safe and protective vaccine is warranted.

Our VLP vaccine provides a safe and feasible approach to address the need for a vaccine against this future threat. The current studies suggest nAb titers induced by the VLP vaccine were consistently high and correlated with protection in both the murine and NHP models. In

our previous work [36], we demonstrated that passive transfer of immune sera from ZIKV VLP-vaccinated mice was sufficient to protect against ZIKV challenge, supporting use of Abs as a mechanism or marker of protection. In general, VLP vaccines have been approved for use in humans with additional candidates in advanced development. Considered together, these observations support the continued development of the VLP approach to prevent ZIKV infection and disease.

## Supporting information

**S1 Data. Complete dataset for all plots and statistical analyses.** Excel.xlsx file with a tab named for each figure panel (e.g. Fig 2B) that contains the corresponding dataset for that panel.
(XLSX)

## Acknowledgments

We also thank BEI Resources, NIAID, NIH for the following reagent: ZIKV, PRVABC59, NR-50240.

## Author Contributions

**Conceptualization:** Lo Vang, Jason Mendy, Darly Manayani, Justin Julander, Daniel Sanford, Jonathan Smith, Jeff Alexander.

**Formal analysis:** Lo Vang, Christopher S. Morello, Justin Julander, Daniel Sanford, Jeff Alexander.

**Investigation:** Lo Vang, Jason Mendy, Danielle Thompson, Darly Manayani, Ben Guenther, Justin Julander, Daniel Sanford, Amit Jain, Jeff Alexander.

**Methodology:** Lo Vang, Jason Mendy, Ben Guenther, Justin Julander, Daniel Sanford, Amit Jain.

**Project administration:** Justin Julander, Daniel Sanford, Paul Shabram, Jonathan Smith, Jeff Alexander.

**Resources:** Justin Julander, Daniel Sanford, Paul Shabram, Jeff Alexander.

**Supervision:** Lo Vang, Jason Mendy, Darly Manayani, Justin Julander, Daniel Sanford, Amish Patel, Paul Shabram, Jonathan Smith, Jeff Alexander.

**Writing – original draft:** Lo Vang, Christopher S. Morello, Jeff Alexander.

**Writing – review & editing:** Lo Vang, Christopher S. Morello, Jason Mendy, Danielle Thompson, Darly Manayani, Ben Guenther, Justin Julander, Daniel Sanford, Amit Jain, Amish Patel, Paul Shabram, Jonathan Smith, Jeff Alexander.

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
