## [Decision Letter · Decision Letter 0]

30 Nov 2020

Dear Mr. Vang,

Thank you very much for submitting your manuscript "Zika virus-like particle vaccine protects AG129 mice and rhesus macaques against Zika virus" for consideration at PLOS Neglected Tropical Diseases. As with all papers reviewed by the journal, your manuscript was reviewed by members of the editorial board and by several independent reviewers. The reviewers appreciated the attention to an important topic. Based on the reviews, we are likely to accept this manuscript for publication, providing that you modify the manuscript according to the review recommendations. 

Sincerely,

Michael R Holbrook, PhD

Associate Editor

Rebecca Rico-Hesse

Deputy Editor

Reviewer's Responses to Questions

**Key Review Criteria Required for Acceptance?**

**Methods**

-Are the objectives of the study clearly articulated with a clear testable hypothesis stated?

-Is the study design appropriate to address the stated objectives?

-Is the population clearly described and appropriate for the hypothesis being tested?

-Is the sample size sufficient to ensure adequate power to address the hypothesis being tested?

-Were correct statistical analysis used to support conclusions?

-Are there concerns about ethical or regulatory requirements being met?

Reviewer #1: The manuscript describes the efficacy of a Zika virus (ZIKV) virus-like particle (VLP) vaccine in AG129 mice and non-human primate. The authors follow the generation of neutralizing antibody using multiple assays and RNAemia following challenge. The methods outlined in the mansucript are acceptable and clearly defined. 

Minor comments-

1. Please define the contents of the “formulation buffer” on line 125.

2. Please provide an explanation for using different alum doses for the non-human primate studies, since the dose of alum remained consistent for each VLP dose in the mouse studies.

Reviewer #2: The Methods section is cluttered and includes redundancies, such as the information about the macaque cohort. 

The use of multiple methods to measure neutralization is impressive.

Reviewer #3: Yes. But methods section needs to be written more concisely and with more clarity. Furthermore statistical analysis of different sample groups need to be done, not just compared to mock (alum groups)

**Results**

-Does the analysis presented match the analysis plan?

-Are the results clearly and completely presented?

-Are the figures (Tables, Images) of sufficient quality for clarity?

Reviewer #1: The results are easy to follow and match the analysis. The figures are clearly presented.

Reviewer #2: Minor modifications requested:

- Does the isolate of PRVABC59 contain the variants known to be associated with attenuation (Duggal et al, https://doi.org/10.1016/j.virol.2019.02.004)? if so, the authors should cite this paper and state the caveats associated with this virus. The data are still valid as this isolate has been used extensively but this information is important for interpretation. 

- some figures represent data post-vaccination but are focused on data post challenge. this is fine, but could be clarified by, for instance, showing the number of days post challenge in parentheses. This is clear in the text, but the figures as they stand are a bit confusing.

Reviewer #3: yes. see attached file for detailed comments.

**Conclusions**

-Are the conclusions supported by the data presented?

-Are the limitations of analysis clearly described?

-Do the authors discuss how these data can be helpful to advance our understanding of the topic under study?

-Is public health relevance addressed?

Reviewer #1: The data presented supports the conclusions of the manuscript.

Reviewer #2: -The authors state the dose used in the macaque studies (~10,000pfu) could be too low or the species not ideal resulting in inconsistent detection of infectious virus. The discussion would benefit from comparison with published studies as various viral inoculum doses and macaque species (e.g. cynomolgus, pigtail) have been used.

Reviewer #3: yes

**Editorial and Data Presentation Modifications?**

Reviewer #1: Minor comments-

1. In all the figures related to the mouse studies, the 0.1ug VLP + alum group is missing the word “VLP”.

Reviewer #2: (No Response)

Reviewer #3: (No Response)

**Summary and General Comments**

Reviewer #1: The study is well written and thought out. While other VLP based vaccines against ZIKV have been tested before in mice, this seems to be the first report in non-human primates. The authors missed an opportunity to show that the antibodies generated following vaccination with this VLP scheme are unlikely to promote antibody-dependent enhancement (ADE), which will be important prior to moving into human trials. Overall, the data supports the conclusions of the manuscript. 

General Comments and Experimental suggestions-

1. Are the structural proteins on the virus-like particles produced fully mature or are there a mixture of mature and immature particles? Please discuss. 

2. The vaccination schemes generated robust neutralizing antibody responses. It would be interesting to know if the majority of the antibodies generated were type specific and/or bound to domain III of the E protein. 

3. Since ADE is of concern particularly in areas with DENV co-circulation, it will be important to show the absence or limited generation of the cross-reactive antibodies or lack of cross-neutralization with DENV.

Reviewer #2: This study is thorough and the manuscript is well-written. Despite the decrease in ZIKV incidence the virus still represents a potential public health threat and vaccines may be needed to stem the spread of a new epidemic should it arise. The use of several methods to measure neutralizing antibodies is impressive and not typical for comparable studies. The use of both mice and primates is likewise impressive. The use of a virus now thought to harbor tissue culture adaptations that result in some in vivo attenuation is unfortunate (if true) but not restricted to this study. However, this information should be included for context as the field moves forward with vaccines and therapeutics.

Reviewer #3: see attached document

PLOS authors have the option to publish the peer review history of their article (what does this mean?). If published, this will include your full peer review and any attached files.

Reviewer #1: No

Reviewer #2: No

Reviewer #3: No
---

## [Decision Letter · Decision Letter 1]

2 Feb 2021

Dear Mr. Vang,

We are pleased to inform you that your manuscript 'Zika virus-like particle vaccine protects AG129 mice and rhesus macaques against Zika virus' has been provisionally accepted for publication in PLOS Neglected Tropical Diseases.

Best regards,

Michael R Holbrook, PhD

Associate Editor

Rebecca Rico-Hesse

Deputy Editor

Reviewer's Responses to Questions

**Key Review Criteria Required for Acceptance?**

**Methods**

-Are the objectives of the study clearly articulated with a clear testable hypothesis stated?

-Is the study design appropriate to address the stated objectives?

-Is the population clearly described and appropriate for the hypothesis being tested?

-Is the sample size sufficient to ensure adequate power to address the hypothesis being tested?

-Were correct statistical analysis used to support conclusions?

-Are there concerns about ethical or regulatory requirements being met?

Reviewer #1: The authors have addressed the concerns of the reviewers.

Reviewer #2: (No Response)

Reviewer #3: Yes

**Results**

-Does the analysis presented match the analysis plan?

-Are the results clearly and completely presented?

-Are the figures (Tables, Images) of sufficient quality for clarity?

Reviewer #1: The authors have addressed the concerns of the reviewers.

Reviewer #2: (No Response)

Reviewer #3: Yes

**Conclusions**

-Are the conclusions supported by the data presented?

-Are the limitations of analysis clearly described?

-Do the authors discuss how these data can be helpful to advance our understanding of the topic under study?

-Is public health relevance addressed?

Reviewer #1: The authors have addressed the concerns of the reviewers.

Reviewer #2: (No Response)

Reviewer #3: yes

**Editorial and Data Presentation Modifications?**

Reviewer #1: (No Response)

Reviewer #2: (No Response)

Reviewer #3: (No Response)

**Summary and General Comments**

Reviewer #1: (No Response)

Reviewer #2: Thank you to the authors for carefully addressing my comments and concerns. All concerns raised by this reviewer were addressed.

Reviewer #3: (No Response)

PLOS authors have the option to publish the peer review history of their article (what does this mean?). If published, this will include your full peer review and any attached files.

Reviewer #1: No

Reviewer #2: No

Reviewer #3: No

---

## [Editor Report · Acceptance letter]

9 Mar 2021

Dear Mr. Vang,

We are delighted to inform you that your manuscript, "Zika virus-like particle vaccine protects AG129 mice and rhesus macaques against Zika virus," has been formally accepted for publication in PLOS Neglected Tropical Diseases.

Best regards,

Shaden Kamhawi

co-Editor-in-Chief

Paul Brindley

co-Editor-in-Chief
